# Regulatory Mechanisms of Exogenous Acyl-Homoserine Lactones in the Aerobic Ammonia Oxidation Process Under Stress Conditions

**DOI:** 10.3390/microorganisms13030663

**Published:** 2025-03-14

**Authors:** Chen Qiu, Kailing Pan, Yuxuan Wei, Xiaolin Zhou, Qingxian Su, Xuejun Bi, Howyong Ng

**Affiliations:** 1State and Local Joint Engineering Research Centre of Urban Wastewater Treatment and Reclamation in China, Qingdao University of Technology, Qingdao 266033, China; qiuchenty0@foxmail.com (C.Q.); pankailing@qut.edu.cn (K.P.); weiyuxuan0119@163.com (Y.W.); zhouxiaolin2680@163.com (X.Z.); 2Centre for Water Research, Advanced Institute of Natural Sciences, Beijing Normal University at Zhuhai, Zhuhai 519087, China

**Keywords:** quorum sensing, activated sludge, ammonia-oxidizing archaea, ammonia-oxidizing bacteria, comammox, ammonia removal, ammonia oxidation inhibitors, community composition

## Abstract

This study investigated the mechanism by which N-acyl-homoserine lactone (AHL) signaling molecules influence ammonia-oxidizing microorganisms (AOMs) under inhibitory conditions. In laboratory-scale sequential batch reactors (SBRs), the effects of different AHLs (C6-HSL and C8-HSL) on the metabolic activity, microbial community structure, and quorum sensing (QS) system response of AOMs were examined. Caffeic acid, 1-octyne, and allylthiourea were used as ammoxidation inhibitors. The results indicated that under inhibitory conditions, AHLs effectively reduced the loss of ammonia oxidation activity and enhanced the resistance of AOMs to unfavorable environments. Additionally, AHLs enriched AOMs in the microbial community, wherein C6-HSL significantly increased the abundance of *amoA* genes in AOMs. Furthermore, AHLs maintained the activity of QS-related genes and preserved the communication ability between microorganisms. Correlation analysis revealed a positive relationship between AOMs and QS functional bacteria, suggesting that AHLs can effectively regulate the ammonia oxidation process. Overall, exogenous AHLs can improve the metabolic activity and competitive survival of AOMs under inhibitory conditions.

## 1. Introduction

Nitrogen pollution in water has become a major global environmental issue, severely affecting water quality, ecosystem stability, and human health [1,2]. Both conventional nitrifying-denitrification processes and unconventional nitrogen removal processes (such as SHARON, CANON, etc.) are inseparable from the conversion of NH_4_^+^. As a critical step in the nitrogen cycle, ammonia oxidation is essential for maintaining the nitrogen balance in Earth’s ecosystems, supporting the growth and reproduction of organisms, and ensuring stable ecosystem function [3,4,5]. In wastewater treatment, activated sludge consists of a complex microbial community that includes various microorganisms involved in ammonia oxidation [6]. Among these, ammonia-oxidizing microorganisms (AOMs)—including ammonia-oxidizing archaea (AOA), ammonia-oxidizing bacteria (AOB), and complete nitrifying bacteria (Comammox)—are the primary drivers of biological nitrogen removal. The community structure, abundance, and activity of these microorganisms are influenced by environmental factors such as the temperature and nutrient load, which, in turn, significantly affect the efficiency of biological nitrogen removal systems [7,8]. However, understanding how these microorganisms respond to environmental changes and how to regulate their activity and abundance remains a significant challenge.

AOMs have been extensively investigated through pharmacological inhibition approaches, establishing chemical suppression as a critical methodology for elucidating microbial nitrogen transformation mechanisms [9,10,11,12,13]. Given that inhibitor application induces functional impairment in sludge consortia, developing targeted reactivation strategies under inhibited conditions becomes imperative for microbial community restoration. The current pharmacological toolkit for ammonia oxidation studies comprises several well-characterized compounds: acetylene (C_2_H_2_), 1-octyne, streptomycin, dicyandiamide (DCD), 3,4-dimethylpyrazole phosphate (DMPP), allylthiourea (ATU), and caffeic acid derivatives [14,15,16]. Mechanistic investigations have quantified inhibitor efficacy through dose–response analyses. Ginestet et al. [17] established that ATU concentrations ≤100 μM achieve complete suppression of ammonia-oxidizing bacteria (AOB) activity. Complementary research by Laura et al. [14] demonstrated equivalent inhibitory potency between 100 μM caffeic acid and 100 μM PTIO in completely arresting archaeal ammonia oxidation (AOA) processes. Differential inhibition strategies have been successfully implemented, as evidenced by Taylor et al. [18] employing 0.03% *v*/*v* 1-octyne to functionally distinguish AOA and AOB contributions in complex microbial systems. Recent molecular investigations by Guo et al. [19] revealed that DMPP exposure significantly downregulates ammonia monooxygenase gene expression (*amoA*), providing genomic evidence for its mode of action.

In recent years, quorum sensing (QS), a microbial communication mechanism mediated by signaling molecules, has garnered significant attention in microbial ecology [20,21]. Acyl-homoserine lactones (AHLs) are the most extensively studied QS molecules in Gram-negative bacteria and play crucial roles in regulating microbial behaviors such as biofilm formation, pathogenicity, and nutrient cycling [22,23]. The critical role of AHL-mediated QS mechanisms in wastewater treatment bioprocesses has been extensively documented through empirical investigations [24]. Notably, Tang et al. [25] detected the secretion of three distinct AHL variants in anammox consortia, with C6-HSL and C8-HSL demonstrating significant enhancement of anammox bacterial activity, while C12-HSL exerted a pronounced stimulatory effect on heterotrophic bacterial proliferation within the microbial consortium. Subsequent analytical characterization confirmed the persistent presence of both C6-HSL and C8-HSL in mature anammox biofilm matrices. Microbial community analyses have identified phylogenetically diverse biofilm-associated species harboring AHL synthase genes, including but not limited to *Pseudomonas*, *Stenotrophomona*, and *Nitrosomonas*. Comparative genomic studies reveal that approximately 3% of isolated bacterial strains possess functional quorum quenching (QQ) genetic determinants [26]. In complementary research, Feng et al. [27] successfully isolated and characterized four AHL homologs (C4-HSL, C6-HSL, C8-HSL, and 3-oxo-C8-HSL) from both activated sludge systems and hybrid biofilm reactors, confirming the ubiquity of QS signaling molecules across different wastewater treatment configurations. At present, most of the existing studies focus on the relationship between quorum sensing and biofilm formation and the dynamic analysis of endogenous signal molecules during nitrogen removal. The research of AHLs, which plays a key role in the regulation of ammonia oxidation, is still in its infancy.

Most AOMs exhibit QS effects in both pure and mixed cultures, and their association with QS has been confirmed through gene sequencing and other techniques [28]. Li et al. [29] found that adding exogenous AHLs could promote the secretion of extracellular polymeric substances, enhance the adhesion of nitrifying sludge, induce the release of endogenous AHLs, and improve microbial resilience under harsh environmental conditions. Clippeleir et al. [30] discovered that exogenous AHL application could boost anaerobic ammonia oxidation rates (AORs) at lower biomass levels. Wang et al. [31] conducted a critical analysis of the fundamental mechanisms and ecological implications of QS in microbial nitrogen transformation cascades, presenting a comprehensive interdisciplinary synthesis of QS-mediated regulatory architectures governing both nitrification and denitrification pathways. Complementary research by Li et al. [29] establishes empirical evidence for the substantive influence of AHLs in facilitating sludge granulation dynamics and associated metabolic functions. Notwithstanding these advances in QS research across diverse microbial consortia, current scientific understanding reveals a paucity of investigations systematically examining the stimulatory effects of AHLs on ammonia oxidation kinetics. Particularly, the molecular governance mechanisms of QS systems during ammonia oxidation processes and their interactions with AOMs remain insufficiently characterized, representing critical knowledge gaps in microbial nitrogen cycle regulation.

This study introduced various inhibitory substances to create stress conditions for the target AOMs and evaluated the impact of adding two AHLs—N-hexanoyl-L-homoserine lactone (C6-HSL) and N-octanoyl-L-homoserine lactone (C8-HSL)—on different AOMs. In addition, microbial community structure and changes in AHL concentrations were analyzed to elucidate the role of AHL-mediated quorum sensing (QS) systems in the ammonia oxidation process. The findings of this study clarified the impact of AHLs on ammonia removal by AOMs and proposed strategies for enhancing ammonia oxidation activity under stress conditions. This study challenges the long-held notion of absolute inhibitor dominance over AOMs and demonstrates that AHL-mediated quorum sensing confers resistance in AOMs against the conventionally perceived suppression of ammonia oxidation by inhibitory agents.

## 2. Materials and Methods

### 2.1. Reactor Operation and Experimental Design

Activated sludge was obtained from the return sludge of the secondary biological treatment system at a sewage treatment plant in Qingdao (China). A specific amount of activated sludge was placed into 600 mL sequential batch reactors (SBRs), and the concentration of mixed-liquor suspended solids (MLSSs) was maintained at 3000 mg/L. Three different inhibitors were added to separate SBRs to selectively inhibit the activity of AOA, AOB, and Comammox. Previous studies have indicated that 100 μM caffeic acid can inhibit AOA metabolism [14], AOB is sensitive to 1-octyne, and 0.03% *v*/*v* 1-octyne effectively inhibits AOB [18], while ATU selectively inhibits the AOB community [17]. To evaluate the impact of AHLs on AOMs, two AHLs (50 μM C6-HSL and 50 μM C8-HSL) were used [32]. The AHL concentrations were established based on precedents in the literature, with deliberately chosen values at the lower spectrum of reported ranges to maintain their functional efficacy in mediating the reaction while minimizing potential confounding effects [33,34,35,36]. The acyl-homoserine lactone AHLs and inhibitors (caffeic acid, 1-octyne, and ATU) were purchased from Sigma-Aldrich.

Table 1 lists the different SBR setups, including the AHLs and inhibitors added and their respective dosages. Three parallel reactors were set up for each experimental objective, totaling 24 SBRs. All reactors treated synthetic domestic wastewater, with its composition and trace elements detailed in Appendix A. The NH_4_^+^-N concentration in the synthetic domestic wastewater was maintained at approximately 56 mg/L. The dissolved oxygen (DO) levels in the reactors were kept at 6–8 mg/L, with a constant temperature of 24 °C. Each SBR operated continuously for 10 reaction cycles.

The concentrations of nitrogen compounds (NH_4_^+^-N, NO_2_^−^-N, and NO_3_^−^-N) and MLSS were measured according to standard protocols (APHA, 2005). The pH and DO levels were monitored using a pH/DO meter (Multi 3420; WTW GmbH, Shenzhen, China). The AOR was calculated based on NH_4_^+^-N removal efficiency per unit MLSS per unit time, expressed as “mg NH_4_^+^-N·h^−1^·g^−1^ MLSS”. After completing 10 cycles, sludge samples were collected from each reactor for microbial community structure analysis.

### 2.2. Illumina MiSeq Sequencing and Sequence Processing

Sludge samples were processed and analyzed to determine bacterial and archaeal microbial diversity using the Illumina MiSeq platform. Bacterial diversity was assessed through polymerase chain reaction (PCR) amplification of reverse-transcribed cDNA using the universal primers 338F and 806R, targeting the V3–V4 hypervariable region of the 16S rRNA gene. This analysis was conducted at Shanghai Majorbio Bio-pharm Technology Co., Ltd. (Shanghai, China).

Archaeal diversity was evaluated using PCR amplification of reverse-transcribed cDNA with the universal primers Arch349F and Arch915R, targeting the V4 hypervariable region of the 16S rRNA gene. This analysis was performed at Beijing Allwegene Tech. Co., Ltd. (Beijing, China).

### 2.3. qPCR Assay of amoA Genes

The *amoA* genes of AOMs were analyzed using real-time quantitative PCR (qPCR) on the ABI QuantStudio™ 1 System (Thermo Fisher, Waltham, MA, USA). The following specific primers were used for qPCR:-AOA *amoA* genes: archaea-amoA F/R (Francis et al., 2005);-AOB *amoA* genes: amoA-1F/1R (Rotthauwe et al., 1997);-Comammox *amoA* genes: com amoA AF/SR (Shao and Wu, 2021).

Each PCR reaction mixture (20 μL) contained 10 μL of SYBR Green Master Mix, 0.4 μL of Rox II, 0.3 μL of each forward and reverse primer, 2 μL of template DNA, and 7 μL of ddH_2_O. A dilution series of plasmids containing the target gene insert (Sangon, Shanghai, China) was used as the standard.

## 3. Results and Discussion

### 3.1. Effects of AHLs on Ammonia Oxidization

The NH_4_^+^-N concentrations in the influent and effluent of each reaction cycle were measured to evaluate the ammonia oxidation activity during the SBR cultivation process. The AOR for each reactor was calculated based on these measurements, as illustrated in Figure 1a.

In Group 1, the AOR of the C6 reactor remained relatively stable throughout the operation, with an average AOR of 1.13 mg NH_4_^+^-N·h^−1^·g^−1^ MLSS over 10 cycles. In the C6-CA reactor, where caffeic acid was introduced to inhibit AOA activity, the AOR decreased slightly during the first cycle but maintained a similar ammonia nitrogen removal efficiency to C6 from cycles 2 to 6 (Figure 1b). However, from cycles 7 to 9, the AOR continuously decreased, showing a 6.88% reduction in the ninth cycle compared to C6. By the 10th cycle, the AOR returned to normal levels.

Under the influence of 1-octyne, the C6-OCT reactor exhibited a 33.05% inhibition effect on AOR during the first cycle. This inhibition gradually diminished over subsequent cycles, decreasing to 22.97% by the third cycle. As the experiment progressed, the AOR in C6-OCT continued to decline, with the strongest inhibition observed in the ninth cycle, where the AOR was reduced by 40.50% compared to C6. The average AOR for C6-OCT throughout the experiment was 0.80 mg NH_4_^+^-N·h^−1^·g^−1^ MLSS. In the C6-ATU reactor, due to the inhibitory effect of ATU, the AOR remained consistently low, with an average value of 0.41 mg NH_4_^+^-N·h^−1^·g^−1^ MLSS, showing no significant fluctuations.

In Group 2, the AOR in the C8 reactor exhibited slight fluctuations and a decreasing trend (Figure 1a), dropping from an initial value of 1.15 mg NH_4_^+^-N·h^−1^·g^−1^ MLSS to 1.12 mg NH_4_^+^-N·h^−1^·g^−1^ MLSS over the course of the experiment. Compared to C8, the AOR in C8-CA showed a gradual increase. During the first four cycles, the AOR of C8-CA remained consistently lower than that of C8; however, from the fifth cycle onward, it began to rise. The most significant increase occurred in the eighth cycle, with an 8.55% improvement relative to C8. After the eighth cycle, the AOR of C8-CA began to decline, returning to 1.12 mg NH_4_^+^-N·h^−1^·g^−1^ MLSS by the final cycle.

The AOR trend in C8-OCT was similar to that of C6-OCT in Group 1, with some fluctuations before the fourth cycle. From the fourth through eighth cycles, the inhibition effect stabilized between 24.19 and 26.66%. However, during the ninth cycle, the inhibition effect increased, leading to an overall inhibition range of 23.23 to 36.94% throughout the experiment. The average AOR for C8-OCT over 10 cycles was 0.80 mg NH_4_^+^-N·h^−1^·g^−1^ MLSS. The addition of ATU also inhibited ammonia oxidation in C8-ATU, with the inhibition effect remaining relatively stable between 66.55 and 67.54% by the final cycle. The NO_2_^−^-N and NO_3_^−^-N concentrations as illustrated in Appendix A.

In the full-cycle monitoring of the first and last cycles (Figure 1c), changes were observed in the NH_4_^+^-N degradation pattern. During the initial full cycle of NH_4_^+^-N degradation across the eight reactors across in both groups, all reactors initially exhibited a higher AOR, which decreased after approximately 30 min. In contrast, during the final cycle of NH_4_^+^-N degradation, the NH_4_^+^-N concentration declined steadily, and all reactors maintained a stable AOR throughout the entire cycle.

In this study, the addition of C6-HSL and C8-HSL did not significantly affect the AOR in the control reactors. Gao et al. [37] reported that short-chain AHLs (C6-HSL and 3-oxo-C6-HSL) had no significant effect on ammonia oxidation after 4 and 14 days of treatment. Similarly, Hu et al. [38] found that a 50 nM mixture of AHLs (C6-HSL, C8-HSL, 3-oxo-C12-HSL, and C14-HSL) did not positively impact ammonia oxidation. However, other studies have shown that C6-HSL and C8-HSL can accelerate NH_4_^+^-N removal to some extent [29,39].

Given the complexity of microbial communities in wastewater treatment, with their intricate QS systems, it is expected that AHLs will have varying effects on the ammonia oxidation process across different reactor systems. In the present study, the addition of inhibitors had diverse effects on NH_4_^+^-N removal in the reactors. The introduction of exogenous caffeic acid did not lead to a significant decrease in the AOR, whereas the addition of 1-octyne reduced NH_4_^+^-N degradation by approximately 30%. Notably, in reactors with continuous ATU addition, C6-ATU and C8-ATU maintained average AOR values of 0.41 mg NH_4_^+^-N·h^−1^·g^−1^ MLSS and 0.38 mg NH_4_^+^-N·h^−1^·g^−1^ MLSS, respectively. Clearly, the suppression of ammonia oxidation by these inhibitors did not reach the levels reported in previous studies [14,17,18]. These results suggested that the introduction of exogenous AHLs influenced microbial activities and impacted NH_4_^+^-N removal. Further analysis of AOM variations in each reactor is warranted to explore how AHLs contribute to maintaining NH_4_^+^-N removal under unfavorable conditions with inhibitors.

### 3.2. Functional Gene Abundance and AOM Diversity

RNA extracted from the activated sludge samples was reverse-transcribed into cDNA to further determine the composition of active AOMs in the sludge. Subsequently, qPCR was employed to quantitatively analyze the *amoA* functional genes of *archaea* and *bacteria*. In Group 1, AOA were the predominant AOMs in three of the reactors, except for C6-ATU. The abundance of AOA *amoA* genes in the four reactors was as follows: C6 contained 6.49 × 10^2^ copies/μL cDNA, C6-ATU contained 5.40 × 10^3^ copies/μL cDNA, C6-CA contained 1.00 × 10^3^ copies/μL cDNA, and C6-OCT contained 6.76 × 10^2^ copies/μL cDNA (Figure 2a). The abundance of AOA *amoA* genes in the inhibitor-added reactors was higher than in C6. In C6-ATU, the abundance of AOB *amoA* genes reached 5.75 × 10^5^ copies/μL cDNA, which was 2111.21 times higher than the second-highest abundance observed in C6-OCT. The abundance of AOB *amoA* genes in both C6-ATU and C6-OCT was higher than in C6. Additionally, C6-ATU exhibited a high abundance of Comammox *amoA* genes, reaching 2.06 × 10^2^ copies/μL cDNA, which was 47.60 times higher than in C6.

In Group 2, where C8-HSL was applied as an exogenous AHL, AOA *amoA* genes predominated. The average abundance of AOA *amoA* genes in all reactors exceeded 1.48 × 10^3^ copies/μL cDNA, with C8 showing the highest AOA *amoA* abundance at 2.93 × 10^3^ copies/μL cDNA. In C8-CA, the transcriptional activity of AOA was significantly suppressed, resulting in an average abundance that was only 57.66% of that in C8. However, the abundance of Comammox *amoA* genes rose by 4.54% compared to C8. In C8-OCT, the abundance of AOB *amoA* genes decreased by 74.83%, while the abundance of Comammox *amoA* genes increased 3.6-fold. In C8-ATU, the abundances of AOA *amoA* genes and Comammox *amoA* genes decreased by 49.59 and 30.87%, respectively, while the abundance of AOB *amoA* genes increased by 4.67% compared to C8.

High-throughput sequencing using Illumina MiSeq was performed on both *archaea* and *bacteria* to comprehensively analyze the functional bacterial evolution following AHL addition. The main AOMs at the species level are shown in Figure 2c. The results revealed that *Nitrososphaera viennensis* and *Candidatus Nitrososphaera gargensis* were the dominant AOA in this study. *Nitrososphaera viennensis* exhibited higher abundance in C6-CA and C6-ATU, whereas *Candidatus Nitrososphaera gargensis* was specifically enriched in C8.

A total of seven AOB species were detected in the experimental samples, with *Nitrosomonas europaea* being the most abundant, comprising over 67.18% in C8-ATU. The overall abundance of AOB was higher in reactors without added inhibitors (8.37% in C6 and 6.67% in C8), while both caffeic acid and ATU significantly reduced the growth advantage of AOB.

In Group 1, C6 significantly enriched all detected AOB species, whereas ATU addition markedly suppressed AOB growth. In C6-OCT, the growth of *Nitrosomonas nitrosa* was significantly stimulated, while the growth of other AOB species was inhibited. Similarly, in C8-OCT, all AOB species except *Nitrosomonas aestuarii* were significantly inhibited. Only one Comammox species, *Candidatus Nitrospira nitrosa*, was detected, and its abundance showed a slight increase in C8.

Based on the community analysis of AOMs, it can be inferred that C6-HSL and C8-HSL may selectively influence different target AOMs in activated sludge. In the presence of inhibitors, C6-HSL more effectively impacted the suppressed target AOMs by enhancing the expression of *amoA* activity genes and increasing the abundance of inhibited AOM species, thereby improving their resistance to unfavorable environmental conditions. In contrast, exogenous C8-HSL did not target suppressed AOMs but instead strengthened the biological activity of AOMs unaffected by inhibitors, thereby maintaining the microbial ammonia removal capacity of the reactor.

Previous studies have attempted to explain the effect of AHLs on NH_4_^+^-N removal. Li et al. [29] demonstrated that C6-HSL and 3-oxo-C6-HSL can promote ammonia nitrogen degradation and enhance the competitive advantage of AOB. Gao et al. [37] investigated the effects of C6-HSL and C8-HSL on microbial community structure and AOR in activated sludge and found that AHL addition significantly improved the AOR after 16 days, with a notable increase in AOB *amoA* gene abundance compared to AOA *amoA*. Research also showed that in a denitrification system with nitrification inhibitors (e.g., DCD), a 1 μM mixture of AHLs (C6-HSL and C8-HSL) boosted the abundance of both AOB and NOB, which suggests that nitrifying bacteria may enhance AHL utilization to overcome unfavorable growth conditions [40]. The effects observed in this study may be related to these two mechanisms. However, under optimal growth conditions (21 °C, DO ≥ 3 mg/L), nitrifying bacteria are less dependent on QS, and the addition of AHLs does not significantly affect ammonia removal or the regulation of functional microbes [38], which is consistent with the findings in the control reactors utilized in the this study. In summary, based on previous research and the results of this study, AHLs appear to be more effective in enhancing the ammonia removal capacity under unfavorable growth conditions, thereby increasing the dominance of AOMs in microbial communities.

### 3.3. Impact of AHL Addition on QS/QQ Bacteria and Genes

In the final cycle of this study, four types of AHLs were detected in the sludge samples: C6-HSL, C8-HSL, C10-HSL, and 3-oxo-C12-HSL (Figure 3a). These AHLs have been widely confirmed to be present in biological tanks of sewage treatment plants and other water treatment facilities [41,42]. The results showed that C6-HSL levels remained stable between 1.97 and 2.49 ng/g in both AHL experimental reactors, with no significant difference between Group 1 and Group 2. Although C8-HSL levels declined, they were overall higher in Group 2 than in Group 1. Reactors treated with 1-octyne and ATU in both groups exhibited elevated C8-HSL levels, with the highest concentration of 1509.93 ng/g observed in C8-OCT.

C10-HSL was detected at the lowest level in C8 (14.21 ng/g), whereas higher concentrations were observed in reactors treated with 1-octyne inhibitors in both groups, with C6-OCT and C8-OCT containing 26.11 ng/g and 24.81 ng/g, respectively. Minimal variation was observed in the concentrations of 3-oxo-C12-HSL, with differences of less than 0.5 ng/g across reactors. The lowest values in both groups were found in reactors supplemented with caffeic acid, with concentrations of 2.59 ng/g (C6-CA) and 2.58 ng/g (C8-CA).

Under continuous AHL addition, only C8-HSL remained present at significant levels in the system, while C6-HSL levels remained low in the sludge. This may be attributed to the greater resistance of long-chain AHLs to hydrolysis compared to short-chain AHLs, making them more stable in the solid phase [43].

A literature review on previously identified AHL-producing bacteria (QS bacteria) and AHL-degrading bacteria (QQ bacteria) were conducted to examine changes in key genera involved in AHL-mediated QS [44]. A total of four QS bacteria were detected (Figure 3b), namely, *Nitrosomonas*, *Nitrospira*, *Enterococcus*, and *Thermomonas*, with *Nitrosomonas* and *Nitrospira* being the most abundant. Both genera exhibited similar distribution patterns across the groups, with the highest abundance found in the AHL-only control reactors, followed by the caffeic acid reactors, and the lowest abundance in the 1-octyne and ATU reactors.

*Nitrosomonas* was the only AOB genus detected in this study, while *Nitrospira* belongs to the Comammox group. The presence of ammonia oxidation inhibitors suppressed the growth of QS bacteria to varying degrees. Previous studies have demonstrated that *Nitrosomonas* and *Nitrospira* can provide homologs of LuxI and LuxR and secrete various AHLs [31]. Additionally, the *Thermomonas* genus produces C8-HSL and stimulates the secretion of extracellular polymeric substances [45].

Furthermore, 10 bacterial genera exhibiting both QS and QQ activity (QS/QQ bacteria) were detected (Figure 3c). The four most frequently detected QS/QQ bacteria were *Pseudomonas*, *Acidovorax*, *Novosphingobium*, and *Delftia*. Notable differences in QS/QQ bacterial abundances were observed across reactors: *Sphingomonas* and *Acinetobacter* were predominantly detected in Group 1, whereas *Stenotrophomonas* and *Mesorhizobium* were more abundant in Group 2. The *Mesorhizobium* genome contains both AHL biosynthesis genes (*luxI*) and AHL degradation genes (hydrolases and acylase genes). *Acidovorax* reached its highest abundance in C6-OCT (1.19%), while *Delftia* abundance increased in reactors supplemented with caffeic acid, reaching relative abundances of 17.01 and 13.70% in C6-CA and C8-CA, respectively. Additionally, 13 QQ bacterial genera were detected (Figure 3d), showing higher diversity than QS bacteria, which was consistent with previous studies [44,46]. The total abundance of QQ bacteria was higher in Group 2 than in Group 1. The two most abundant QQ bacteria were *Comamonas* and *Pseudoxanthomonas*. *Sphingopyxis* was more abundant in ATU-treated reactors, while *Cloacibacterium* was specifically detected in reactors containing caffeic acid.

Based on PICRUSt 2 predictions, 35 QS-related functional genes were identified in this study (Figure 3e). Among these, 14 AHL-related genes were retrieved, including 13 QS-related genes (2 AHL biosynthesis genes and 11 AHL-sensing genes) and 1 QQ-related gene (an AHL lactonase gene). In both groups, the abundance of most AHL-QS and AHL-QQ-related genes was higher in the AHL-only control reactors than in those with inhibitors, suggesting that inhibitors suppressed AHL-QS and AHL-QQ activity to some extent. However, certain AHL-QS and AHL-QQ genes in the inhibitor-treated reactors exhibited similar or even higher abundances compared to the AHL-only control reactors.

In Group 1, genes such as *bjaI*, *luxR*, and *cciR* maintained higher abundances across reactors, while *lasR*, *sdiA*, *bjaR1*, *traR*, and *sinR* were more abundant in Group 2 reactors. This indicates that the addition of C6-HSL and C8-HSL in the presence of inhibitors preserved the activity of specific AHL-QS genes, thereby facilitating bacterial communication. This finding suggests that the two AHL-mediated QS pathways function differently, which may also explain their distinct target genera.

### 3.4. Potential Interactions Between AHL-Related Bacteria and the Ammonia Oxidation Process

Figure 4 illustrates the ammonia oxidation process influenced by AHLs, including performance, functional genes, and functional bacterial species, along with potential interactions with QS/QQ-related functional bacteria. A significant positive correlation was observed between QS bacteria and the four AOB species, with the strongest correlation found in *Nitrosomonas europaea* (*p* ≤ 0.001). *Nitrosomonas europaea* was the most abundant AOB in all reactors in this study. Additionally, there was also a notable positive correlation between QS bacteria and AOB *amoA* gene abundance (*p* ≤ 0.05). Most correlations with other AOM species were also positive, leading to a significant positive correlation between QS bacteria and the AOR. This finding suggests that AHL-mediated QS plays a crucial role in the ammonia oxidation process.

Although AHLs have previously been identified as signaling molecules for intraspecies communication, they can also be synthesized and expressed by multiple species, suggesting that AHLs can participate in interspecies communication. Previous studies have shown that AHL synthesis and utilization are widespread among AOB. For example, *Nitrosomonas europaea* can utilize C8-HSL [41], while *Nitrosospira multiformis* and *Nitrosospira briensis* can synthesize various AHLs, including C8-HSL and C10-HSL [47]. In summary, further research is needed to elucidate the mechanisms by which AHL-mediated QS influences interspecies communication under adverse conditions.

## 4. Conclusions

This study found that under inhibitory environmental conditions, the addition of AHLs can regulate the activity and abundance of AOMs in activated sludge. Furthermore, 50 μM C6-HSL and C8-HSL could restore the ammonia oxidation activity of AOMs affected by inhibitors, yet the underlying mechanisms differed. C6-HSL played a crucial role in the ammonia oxidation process and was extensively utilized by suppressed AOMs to adapt to adverse conditions, while C8-HSL stimulated unsuppressed AOMs in the presence of inhibitors, thereby enhancing their ammonia oxidation removal capacity. The increased abundance of QS bacteria and the active expression of AHL-QS-related genes positively regulated the functional recovery and community structure of AOMs under unfavorable conditions. AHL-mediated QS has significant potential for restoring AOM performance and regulating the AOM community structure in stressful environments.

Although exogenous AHLs exhibit promising therapeutic applications for reactivating suppressed AOMs, the structure–function relationships governing congener-specific interactions with ammonia monooxygenase (AMO) catalytic centers demand comprehensive mechanistic investigation through integrative interdisciplinary methodologies employing meta-omics profiling and enzymatic characterization. Given their ubiquitous presence as signaling molecules within wastewater microbiomes, AHLs exert critical regulatory influence on biogeochemical processes spanning nitrogen cycling (particularly the coupling between nitrification and denitrification processes) and recalcitrant pollutant biotransformation (e.g., polycyclic aromatic hydrocarbon catabolism). Systematic investigation of AHL-mediated interkingdom communication systems—particularly their multifaceted regulatory impacts on interconnected metabolic cascades through spatial-temporal modulation—constitutes a pivotal research frontier for advancing environmental biotechnology applications.

## Figures and Tables

**Figure 1 microorganisms-13-00663-f001:**
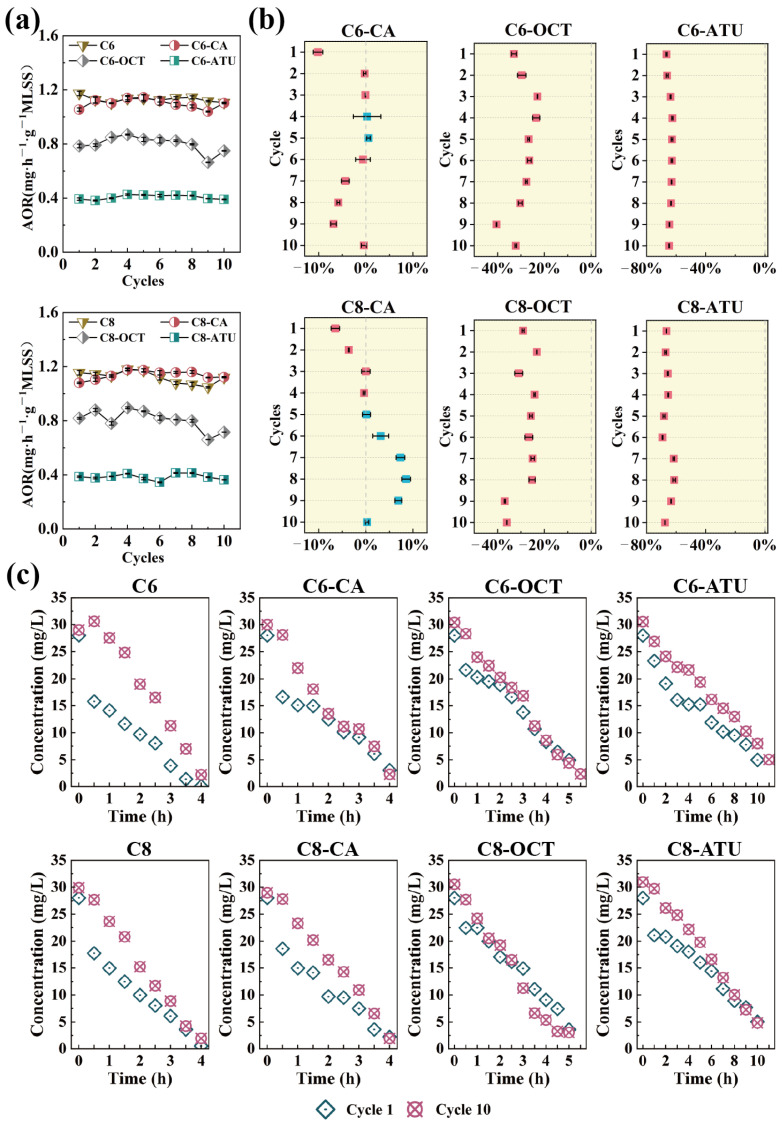
(**a**) Ammonia oxidation rate (AOR) per cycle during the experiment; (**b**) ammonia oxidation inhibition effects in the experimental reactor compared to the control reactor; and (**c**) changes in ammonia–nitrogen processes during the first and last cycles of the experiment.

**Figure 2 microorganisms-13-00663-f002:**
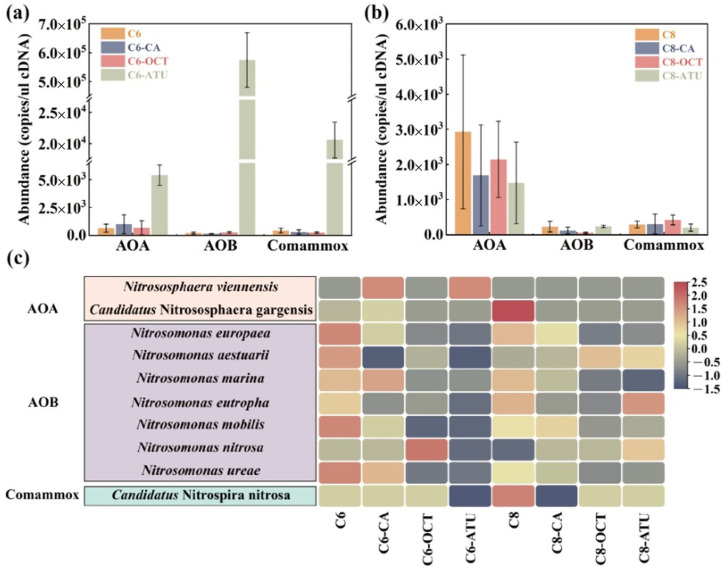
*amoA* gene abundance of ammonia-oxidizing microorganisms (AOMs) in (**a**) Group 1 and (**b**) Group 2; (**c**) species-level ammonia-oxidizing functional bacterial abundance. AOA: ammonia-oxidizing archaea; AOB: ammonia-oxidizing bacteria; Comammox: complete nitrifying bacteria.

**Figure 3 microorganisms-13-00663-f003:**
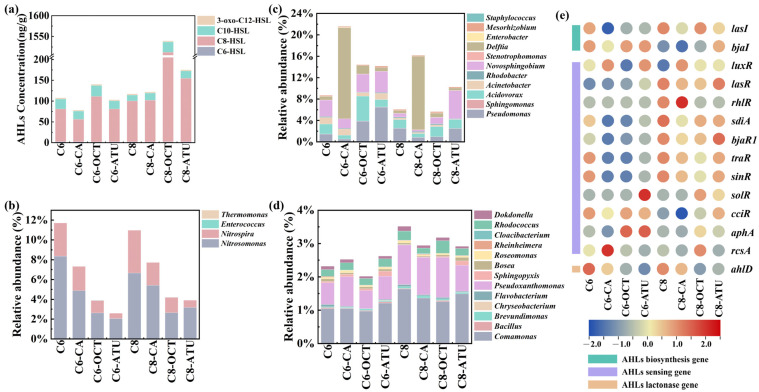
(**a**) Acyl-homoserine lactone (AHL) contents in sludge samples; (**b**) relative abundance dynamics of aggregates and genes associated with AHL-mediated communication: quorum-sensing (QS) bacteria; (**c**) QS/AHL-degrading (QQ) bacteria; (**d**) QQ bacteria; and (**e**) AHL-related functional genes.

**Figure 4 microorganisms-13-00663-f004:**
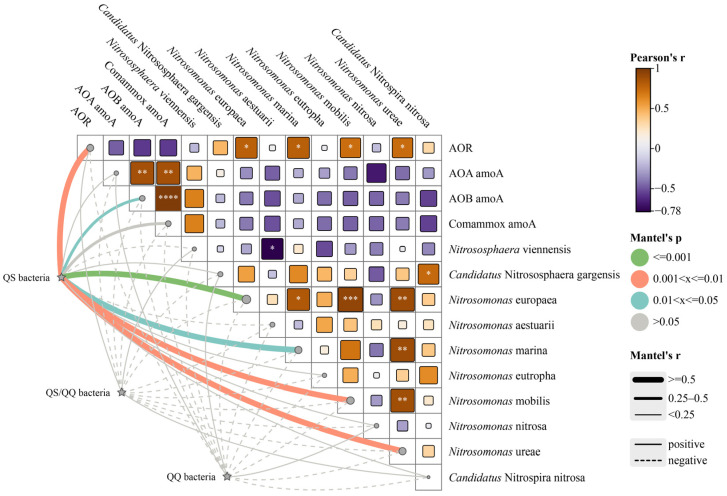
Correlations between the ammonia oxidation rate (AOR), *amoA* genes, ammonia-oxidizing functional species, and community-sensing microorganisms. Asterisk (* or ** or *** or ****) was added if the correlation is significant (*p* < 0.05 or *p* < 0.01 or *p* < 0.001 or *p* < 0.0001, respectively).

**Table 1 microorganisms-13-00663-t001:** AHLs and inhibitors added to SBRs.

Inhibitor and Dose	Blank Control	Caffeic Acid, 100 μm	1-Octyne, 0.03% *v*/*v*	ATU, 100 μm
C6-HSL, 50 μM (GROUP 1)	C6	C6-CA	C6-OCT	C6-ATU
C8-HSL, 50 μM (GROUP 2)	C8	C8-CA	C8-OCT	C8-ATU

## Data Availability

The original contributions presented in this study are included in the article/Appendix A. Further inquiries can be directed to the corresponding author.

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
