# Peer review of "Regulatory Mechanisms of Exogenous Acyl-Homoserine Lactones in the Aerobic Ammonia Oxidation Process Under Stress Conditions"

_microorganisms, 2025, doi:10.3390/microorganisms13030663_

Round 1
Reviewer 1 Report
Comments and Suggestions for Authors
The study addresses a relevant and timely issue in microbial ecology and wastewater treatment, focusing on the role of AHLs in regulating ammonia-oxidizing microorganisms (AOMs) under stress conditions. The experimental design uses sequential batch reactors (SBRs) and a variety of inhibitors to simulate stress conditions.The use of qPCR and Illumina MiSeq sequencing provides comprehensive data on microbial community structure and functional gene abundance.
Comments and suggestions:
1) Abstract and keywords
- The abstract provides a concise summary of the objectives, methods, results, and conclusions. However, it could briefly mention the specific inhibitors used to provide more context.
2) Introduction, novelty and objectives
- The novelty of the research is described, particularly in terms of exploring the differential effects of C6-HSL and C8-HSL on AOMs under stress. However, the manuscript could more explicitly highlight how this study advances beyond previous work in the field.
- The introduction could include more information on the specific inhibitors used (caffeic acid, 1-octyne, and ATU) and their known effects on AOMs. This would help readers understand the rationale behind their selection.
- A discussion of the potential for AHLs to act as signaling molecules in mixed microbial communities, beyond their role in AOMs, would provide a broader perspective.
- Consider adding references to studies that have explored the role of QS in other microbial processes, such as biofilm formation and pathogenicity, to provide a more comprehensive background.
Suggested references to overcome these flaws:
https://doi.org/10.1016/j.chemosphere.2021.129970
https://doi.org/10.1038/npjbiofilms.2015.6
- The introduction lacks information on non-conventional pathways of nitrogen removal that is important to discuss in the scope of the results. Discussing the influence of the C/N ratio and the interrelation between nitrifying, denitrifying, and phosphorus-accumulating microorganisms would emphasize the interconnected nature of microbial processes in wastewater treatment. This is particularly important because AHLs are known to influence not only nitrifiers but also other functional microbial groups.
- By referencing unconventional nitrogen removal mechanisms (e.g., anammox, SHARON, CANON), the authors can better highlight the novelty of their work. For example, while many studies have explored AHLs in conventional nitrification-denitrification, fewer have examined their role in more specialized processes like anammox or partial nitritation.
Suggested references to overcome these flaws:
https://doi.org/10.1038/nature10453
https://doi.org/10.1016/j.biortech.2009.07.016
https://doi.org/10.1128/AEM.02337-09
https://doi.org/10.1016/j.biortech.2014.10.021
3) Material and Methods
- The manuscript does not clearly explain the rationale for selecting the specific AHLs (C6-HSL and C8-HSL) and their concentrations (50 μM). A more detailed justification based on previous studies or preliminary experiments would strengthen the methodology.
- How stable are AHLs in the reactor environment, and could degradation products influence the observed effects?
4) Results and Discussion
- The study lacks a discussion on the potential ecological implications of introducing exogenous AHLs into natural or engineered systems. For example, could this lead to unintended consequences, such as the proliferation of undesirable microbial species?
- The manuscript does not address the potential for AHL degradation or the stability of these molecules in the reactor environment over time, which could influence the observed effects.
- Can the authors elaborate on the mechanisms by which C6-HSL and C8-HSL differentially affect suppressed and unsuppressed AOMs?
- What are the potential ecological risks of introducing exogenous AHLs into wastewater treatment systems?
- How scalable is this approach for real-world wastewater treatment plants, and what are the potential challenges?
- Could the introduction of AHLs lead to the proliferation of undesirable microbial species or the disruption of existing microbial communities?
- Expand the discussion to include the broader implications of the findings for wastewater treatment and microbial ecology. Address the potential ecological risks and challenges associated with introducing exogenous AHLs into natural or engineered systems.
5) Conclusions
- The conclusion that "AHL-mediated QS has significant potential for restoring AOM performance" is somewhat speculative. While the data support the idea that AHLs can enhance AOM activity under stress, the long-term implications and scalability of this approach are not discussed.
- The claim that "C6-HSL and C8-HSL function differently" is supported by the data, but the underlying mechanisms are not fully explored. For example, why does C6-HSL target suppressed AOMs, while C8-HSL stimulates unsuppressed AOMs? This could be further investigated.
- Provide a more detailed discussion of the long-term implications and scalability of the findings.
Reviewer 2 Report
Comments and Suggestions for Authors
The authors set the stage to investigate the mechanism by which N-acyl-homoserine lactone (AHL) signaling molecules influence ammonia-oxidizing microorganisms (AOMs) under inhibitory conditions. For this purpose, they introduced various inhibitory substances (caffeic acid, 1-octane, ATU) to create stress conditions for target AOMs and they evaluated the impact of adding two AHLs on different AOMs. Microbial community structure and changes in AHL concentrations were also analyzed to elucidate the role of AHL-mediated quorum sensing (QS) systems in the ammonia oxidation process. The results revealed that under inhibitory conditions, AHL enriched AOMs in the microbial community, wherein C6-HSL significantly increased the abundance of amoA genes in AOMs and maintained the activity of QS-related genes. Correlation analysis revealed a positive relationship between AOMs and QS functional bacteria, suggesting that AHLs can effectively regulate the ammonia oxidation process. This study is interesting, but there are some issues that have to be addressed.
1) The authors should include more literature studies regarding quorum sensing (QS) mediated by AHL molecules
2) The authors state in introduction that there are extensive studies on QS in various microbial processes (line 66-67). They should justify this sentence with some references.
3) Have the authors investigated the impact of inhibitor concentration on AOR ?
4) The authors state that certain AHL-QS and AHL-QQ genes in the inhibitor-treated reactors exhibited similar or even higher abundances compared to the AHL-only control reactors (line 317-318). Could the authors explain why that happens?
Reviewer 3 Report
Comments and Suggestions for Authors
Dear authors,
The work provides information on the role of AHL under stressful conditions. There are some parts that need inclusion of additional words and here are my suggestions aiming to improve the manuscript:
- Lines 52, and 84: it should be standardized units of concentration. I suggest using µM as appears in line 84 and others.
- Considering suggestion #1, kindly convert 0.03% (v/v) into µM and all the parts it is presented in full manuscript.
- Line 63: include examples of harsh environmental conditions. Include new references and this information is important further in discussion.
- Section 2.1: please provide the ratio active sludge: aqueous phase
- Line 93: provide the synthetic domestic wastewater composition (use text in place of table)
- Table 1: kindly convert MM into µM. The correct term is concentration and not dose. Regarding 1-octyne, see suggestion 2.
- Line 195: this part of the manuscript authors reported genes of prokaryotic domains. So, please use Archaea and Bacteria (both italicized)
- Lines 216-217: same as suggestion 7
- Line 251: what are the optimal conditions? Your work described DO, pH and temperature. It is important to say that reference #30 has being mentioned. So, kindly discuss what optimal conditions mean.
- Section 3.3:
- A) kindly include short lines by discussing how the microbial relation QS/QQ occurs in “natural” high selective pressure environments.
- B) You should define and discuss the concept of public goods and cheating in microbes. C6-AHL is more common and acts interspecies. Is it important for survival in selective environments? The answer completes observation 10a
- Based on your results, include insights into wastewater treatment. It is welcome to provide applications of them (discussion or perspectives in conclusion in further applied studies)
Round 2
Reviewer 1 Report
Comments and Suggestions for Authors
The manuscript has improved somewhat in quality but could be more robust by including more information on unconventional mechanisms of N.
Comments on the Quality of English LanguageEnglish must be reviewed by a native speaker.
Reviewer 2 Report
Comments and Suggestions for Authors
-
Author Response
We sincerely appreciate your continued participation in the peer-review process of our manuscript. Your expertise has been instrumental in maintaining the scholarly rigor of this work.